# A Complete Functional Characterization of Patients with Severe Knee Osteoarthritis in Need of Total Knee Replacement

**DOI:** 10.3390/jfmk9040216

**Published:** 2024-11-01

**Authors:** Vinicius Taboni Lisboa, Bruno de Paula Leite Arruda, Rafael de Andrade Tambascia, Alessandro Rozin Zorzi, Alberto Cliquet, Gustavo Constantino de Campos

**Affiliations:** 1Departamento de Ortopedia, Reumatologia e Traumatologia, Universidade Estadual de Campinas, Campinas 13083-887, Brazil; fisio.viniciustaboni@gmail.com (V.T.L.); arzorzi@unicamp.br (A.R.Z.);; 2Instituto Wilson Mello, Campinas 13080-658, Brazil; bplarruda@gmail.com (B.d.P.L.A.); rafat6@gmail.com (R.d.A.T.)

**Keywords:** knee osteoarthritis, total knee arthroplasty, physical functional performance, quality of life, chronic pain

## Abstract

Background/Objectives: The current literature lacks objective criteria to correctly identify patients in need of a total knee replacement. Surgery indication can be challenging for orthopedic surgeons, which may lead to high levels of patient dissatisfaction. The objective of this study is to describe a complete set of functional characteristics to identify patients with end-stage knee osteoarthritis in need of a total knee replacement, correlating data from strength and performance tests with pain, function, and quality of life questionnaires. Methods: This was a cross-sectional study evaluating patients with end-stage knee osteoarthritis in a waiting list for total knee replacement at a University Hospital. The patients responded to subjective self-reported questionnaires and performance-based functional tests. Anthropometric data were also collected. The main outcome measures were Western Ontario and McMaster Universities Index (WOMAC), visual analog scale (VAS), Short Form-36, knee range of motion, thigh perimeter measurement, maximum voluntary isometric contraction, and 6-min walk test. Results: We analyzed 122 patients (89 female). The functional profile of patients with severe knee osteoarthritis awaiting total knee replacement was described. Quadriceps strength (extensor torque) had a negative correlation with WOMAC (r = −0.3102; *p* < 0.05), VAS (r = −0.3247; *p* < 0.05), and a positive correlation with SF-36 Functional Capacity subscale (r = 0.321; *p* < 0.05). Poorer performance in the 6 min walk test also correlated with worse scores in the WOMAC (r = −0.35; *p* < 0.05), VAS (r = −0.48; *p* < 0.05) and SF-36. Conclusions: The present article established a functional profile of patients with severe knee osteoarthritis with indication for total knee replacement, which may help orthopedic surgeons in their decision process. We also identified quadriceps strength and a 6 min walk test as the two most important functional parameters that correlate with knee osteoarthritis severity.

## 1. Introduction

Knee osteoarthritis (KOA) is a heterogeneous disease characterized by progressive articular cartilage breakdown, subchondral bone remodeling, ligament and meniscal degeneration, and synovial inflammation, along with damage at the molecular and cellular levels. These structural alterations ultimately lead to joint pain, stiffness, and functional disability [1]. The world prevalence of osteoarthritis (OA) is estimated to be about 250 million people, with the knee being the most commonly affected joint, followed by the hand and hip [2]. The prevalence is increasing due to global aging and increasing obesity levels rates. The healthcare cost of osteoarthritis is estimated to be between 1% and 2.5% of the gross domestic product of high-income countries, with joint replacements representing a major proportion of these costs [2]. In the United States, treatment costs for KOA are approximately USD 30 to USD 40 billion per year, covering medical care, medications, physical therapy, and surgeries [3]. In Brazil, the number of OA cases nearly doubled between 2000 and 2017 [4]. Knee osteoarthritis accounts for approximately 85% of the OA burden globally [2].

Most of the KOA treatment guidelines recommend a combination of non-pharmacological and pharmacological therapies, including information/education, weight loss, exercise programs, analgesics, nonsteroidal anti-inflammatory drugs, and symptomatic slow-acting drugs [1]. Muscle strengthening is one of the key treatments, along with weight loss. Studies indicate that combining exercise with weight loss yields better outcomes for pain and function than either approach alone [2]. Surgical treatment has its place only when all other appropriate treatments have failed. Total knee replacement (TKR) still plays an important role in end-stage disease [5].

Patients with KOA experience a significant reduction in daily movements. This pattern of physical hypoactivity leads to muscle weakening, increased joint stiffness, and deconditioning, which contributes to the worsening of the condition [6]. In the United States, it is estimated that approximately 22.5 million adults cannot walk more than three city blocks, and 21.7 million have difficulty climbing stairs [7]. Meta-analyses have shown a significantly increased risk of both diabetes and cardiovascular disease among OA patients, likely due to decreased mobility [8]. Furthermore, OA has also been associated with increased mortality [8]. Thus, the correct identification and treatment of end-stage disease is critical to alleviate the burden of KOA. Selecting the right patients is essential for a successful TKR, as dissatisfaction rates after a primary TKR can be as high as 25% [9]. Various preoperative factors contribute to dissatisfaction, making it important to identify these factors in order to improve patient selection [10]. Currently, no single leading factor has been identified as most predictive of outcomes, and the literature is conflicting in this regard. Higher preoperative function and lower stages of arthritic disease have been associated with dissatisfied patients [10], along with worse preoperative WOMAC scores and older age [11]. Importantly, the relationship between radiographic severity and KOA symptoms is not direct [12], nor is the relationship between radiographic severity and TKR outcomes [11]. Moreover, the current literature poorly correlates preoperative factors for the decision to perform surgery with clinical outcomes and the level of patient satisfaction [11,13]. Some articles describe functional parameters such as quadriceps strength and performance tests in patients with severe KOA and even suggest a functional profile for the patient that indicates TKR [14,15,16]. However, the current literature lacks objective criteria to accurately identify patients in need of total knee replacement, a crucial aspect given that improper indication can lead to high levels of patient dissatisfaction [9,10,17]. Thus, the present article presents a comprehensive set of self-reported and performance-based pain and functional characteristics to identify severe KOA patients needing surgical treatment with TKR, correlating data of strength and performance tests with pain, function, and quality of life questionnaires.

## 2. Materials and Methods

### 2.1. Study Design

This cross-sectional, descriptive, hypothesis-generating study was approved by the local research ethics committee from the State University of Campinas (UNICAMP) under protocol number CAAE 45875015.3.0000.5404 and conforms with the Helsinki Declaration.

### 2.2. Patients

Patients were recruited from a waiting list for TKR at an orthopedic outpatient clinic at the State University of Campinas Clinical Hospital, Campinas, Brazil. Functional and clinical assessment data were collected. The study objectives, benefits, and risks were explained to all patients. Patients who agreed to participate signed an informed consent form. All patients’ data were stored under hospital identification codes, with no personal names or photos associated. Demographic data were collected, including age, weight, height, body mass index [BMI], and duration of symptoms. All patients underwent knee radiographs including anterior to posterior with unilateral weight bearing, lateral, and patellar axial views. Two authors (GCC, ARZ) reviewed the radiographs, classifying OA severity using the Kellgren and Lawrence scheme [18]. In cases of interobserver disagreement, a third opinion was sought. Observers were blind to functional results.

Inclusion criteria were (i) knee pain for at least a year with VAS > 3; (ii) radiographic knee osteoarthritis Kellgren and Lawrence scheme > 2; (iii) patients on a waiting list for a TKR at a University Orthopedics service, as indicated by an experienced surgeon. We excluded patients with signs of knee joint infection, pregnancy, and patients with another disease not correlated with KOA that could interfere with the performance of the functional tests, such as severe respiratory disease or lower limb palsy.

### 2.3. Outcome Measures

Patients were situated in a private room for demographic data collection, questionnaire responses, and functional tests, which took about 60 min. The equipment used included an examination table, a couch, a 360° plastic goniometer to measure range of motion (ROM), a measure tape, a hand-held dynamometer for maximum strength measurements (Lafayette Instrument Company, 01165, Lafayette, IN, USA), inelastic belt straps for stabilizing the dynamometer, a stopwatch, and two cones for the 6-min walk test (6MWT) circuit.

### 2.4. Functional Tests

Knee ROM was measured with a goniometer with the patient in a supine position, with the knee passively moved from full extension to full flexion. Thigh perimeter was measured with a tape at 6 cm and 12 cm above the upper patellar edge. Maximum voluntary isometric contraction of the quadriceps was measured bilaterally. For this test, examiners maintained verbal encouragement during measurements, asking subjects to push as hard as possible against the device. Examiners used both hands to stabilize the device during the test, but no manual resistance against the shin was applied. Patient was positioned sitting by the edge of the stretcher, with both knees and legs out and arms crossed at shoulder height. The knee was positioned at 60 degrees and the dynamometer was positioned at the distal shin (Figure 1).

The test involved two repetitions of maximal strength, each lasting six seconds, with a 20 s rest between repetitions. Results were averaged, normalized by body weight, and defined as quadriceps torque strength.

For the 6MWT, participants walked a course marked by cones 15 m apart for six minutes, with the total distance recorded. The minimum ideal distance was calculated following Enright et al. [19], with the deficit calculated as (Deficit = distance traveled/minimum ideal distance) × 100. The “index knee” was the osteoarthritic knee, or the “worst” knee in cases of bilateral disease.

### 2.5. Self-Reported Questionnaires

Western Ontario and McMaster Universities Index (WOMAC) [20]; visual analog scale (VAS); and 36-Item Short-Form Health Survey (SF-36) [21] were used.

### 2.6. Statistical Analysis

No pre-study sample size calculation was performed, as we recruited all patients on the waiting list. However, a post-study calculation confirmed the adequacy of the sample size. All strength data were normalized for body mass using the following equation: [(kg force/kg body mass) × 100]. Descriptive analysis was performed, with minimum and maximum values, means, standard deviations, and quartiles calculated for quantitative variables, and absolute and relative frequencies calculated for qualitative variables. Spearman’s correlation coefficient was used to study correlations, with calculations performed in SPSS 17.0 for Windows at a significance level of 5%. Only patients with complete data were included in the final analysis, and those with missing data were excluded.

## 3. Results

After inclusion and exclusion criteria, 122 patients were analyzed (89 female) with a mean (SD) age of 68 (8) and a mean BMI of 29.42 kgm^2^ (4.53). The mean length of symptoms was 52 months (50). The demographic characteristics of participants are shown in Table 1.

The functional profile of patients with severe KOA awaiting TKR is described in Table 2.

In our sample, we found a mean knee flexion of 121° (12) in the index knee and 130° (10) in the contralateral knee. The mean extension was −4° (6) in the index knee and −1° (4) in the contralateral knee. Negative values represent a lack of extension. The mean extensor torque was 19.1 kgf (8) in the index knee and 24.3 kgf (8.7) in the contralateral knee (Table 2).

In the 6MWT, patients had an average performance of 318.2 (134.1), with a deficit of −26.64% (30.77) compared to the predicted value for their sex, weight, height, and age. The results of the 6MWT are described in Table 3.

Patients had an average WOMAC score of 46.1 (17.7), while the average score on the VAS was 4.7 (2.8). For the SF-36 questionnaire, we calculated the mean for each subscale, resulting in the following scores: Mental Health 46.1 (17.7), Emotional Aspects 51.1 (39.4), Social Aspects 62.9 (26.1), Vitality 54.8 (23.1), Overall Health Status 61.8 (23.4), Pain 40.3 (22.4), Physical Limitation 32.1 (33.3), and Functional Capacity 29.5 (20.2). The self-reported outcomes are shown in Table 4.

Table 5 shows Spearman’s coefficient for each correlation between two variables, comparing functional data with self-reported questionnaires (WOMAC, VAS, and SF-36). A deficit in knee flexion correlated with poorer performance in the WOMAC questionnaire (r = −0.3296; *p* < 0.05). Quadriceps strength (extensor torque) had a negative correlation with the WOMAC (r = −0.3102; *p* < 0.05), VAS (r = −0.3247; *p* < 0.05) and a positive correlation with the SF-36 Functional Capacity subscale (r = 0.321; *p* < 0.05). Finally, poorer performance in the 6MWT correlated with lower scores in all self-reported questionnaires except for the SF-36 Physical Limitation subscale (Table 5).

In addition to the correlations between functional aspects and questionnaires, we assessed the correlation between thigh circumference (at 6 cm and 12 cm) and the extensor torque. The extensor torque was found to have a significant positive correlation with thigh circumferences at 6 cm (r = −0.4372; *p* < 0.05) and 12 cm (r = −0.4144; *p* < 0.05) (Figure 2 and Figure 3).

## 4. Discussion

The primary goal of this study was to establish a functional profile of patients with severe KOA who were indicated for total knee replacement. To our knowledge, this is the first study to define a complete set of self-reported and performance-based pain and functional characteristics for identifying KOA patients who need TKR. We also identified two key functional parameters associated with severe symptoms: quadriceps strength and 6MWT performance.

Accurate patient selection is crucial for successful TKR. Factors such as patient age, radiographic severity, and symptom severity (including response to other treatment modalities) are widely considered the most important factors that should be taken into account for a surgeon to recommend TKR [11]. However, TKR indication can be challenging for orthopedic surgeons especially when not all those factors are met, leading to high levels of patient dissatisfaction [9,10,17]. A correct alignment of the patient’s expectations is critical, as unmet expectations are a primary cause of dissatisfaction [9,10,17].

The discordance between radiographic changes and knee pain in knee osteoarthritic patients is well established [12]. This phenomenon is very important since radiographic severity plays a major role in a TKR indication decision. Chang et al. found no significant associations between the radiographic severity of OA and the postoperative outcomes of TKA, challenging the notion that performing TKR in patients with lesser radiographic severity results in inferior postoperative outcomes [11]. In our study, the radiographic severity of KOA did not correlate to any other parameter, suggesting that functional parameters should play a larger role in the TKA decision process. Our sample had a decreased ROM, thigh circumference, and quadriceps strength compared to normative values in the literature [22]. A decreased ROM is common in KOA patients, and restricted joint mobility is recognized as an important component of disability in these patients.

Our patients had a mean quadriceps strength of 19.13 kgf, approximately 24% of their body weight. In contrast, Andrew et al. found a normative parameter of 33.65 kgf or 45% of body weight [22]. Several studies have linked quadriceps weakness to both the development and progression of KOA [23,24,25]. A recent systematic review and meta-analysis including 46,819 men and women confirmed knee extensor muscle weakness as a risk factor for the development of KOA [23]. The main explanation for this association is that the quadriceps muscle regulates knee joint loading and motion, which is critical for cartilage homeostasis. A weak quadriceps can negatively influence loading distribution, leading to degenerative joint pathology [23]. Quadriceps strength also predicts knee replacement risk independent of radiographic disease and pain in women [24]. In the present study, quadriceps weakness correlated with poorer outcomes on nearly all subjective scales and functional tests. It is also noteworthy that arthroplasty results tend to be worse in individuals with preoperative quadriceps weakness [25]. Thus, we strongly advocate that patients with KOA, whether undergoing conservative or surgical treatment, prioritize quadriceps strengthening exercises.

The 6MWT was also found to be highly correlated to poor self-reported and performance-based pain and functional outcomes. Therefore, we believe it to be the best performance test alongside with quadriceps strength measurement, as it is a simple, affordable test with normative data available for comparison [19]. Rather than having pain at rest, most of the time, the KOA patient experiences pain during loading movements, which is called movement-evoked pain, a major cause of disability and reduced quality of life in patients with KOA [26]. Unlike other assessments, the 6MWT captures the cumulative impact of KOA on a patient’s ability to perform sustained physical activity in a real-world context. Consequently, this makes the 6MWT an invaluable tool for monitoring OA progression and the need for surgical treatment.

Another important finding was the correlation between thigh circumference (at 6 cm and 12 cm) and the extensor torque. We found a significant positive correlation between the extensor torque and thigh circumference at 6 cm (r = −0.4372; *p* < 0.05) and 12 cm (r = −0.4144; *p* < 0.05). Thigh circumference is a reliable measure of muscle mass and can serve as a proxy to identify sarcopenia, independent of body mass composition or fat distribution [27]. Thus, if it is impossible to measure maximum strength due to the unavailability of a dynamometer, thigh circumference could work as a surrogate outcome, since it is easy to perform and virtually inexpensive.

Our study also showed low scores on the emotional aspects and mental health subscales of the SF-36. Mental health issues are common in KOA patients. A recent systematic review and meta-analysis examined 14 studies involving adult patients with KOA and found a significant prevalence of depression and anxiety among older KOA patients. They found a pooled prevalence of depression of 30% and a combined prevalence of anxiety of 27% [28]. Persistent pain and mobility restrictions are likely major contributors to these mental health conditions. It has been shown that patients with higher painDETECT scores (a questionnaire used to assess the presence of central pain) presented with persistent high pain after treatment [29]. Thus, this could also be used as a screening tool to identify patients who could be dissatisfied after a TKR. Mental health interventions should definitely be integrated into KOA treatment plans. Further research in this area is warranted.

This study has limitations. First, all subjects were referred for KOA treatment at our tertiary care hospital, which may explain the lack of correlation between radiographic severity and other parameters, as most patients had severe radiographic disease. On the other hand, we sought to identify the typical end-stage KOA patient. Second, due to the slowness of our local public health system, patients often presented with prolonged symptoms (averaging 4 years and 4 months), which may negatively impact subjective and functional results. Third, while we assessed mental health using the SF-36, future studies should include more targeted tools for anxiety, depression, and catastrophizing. Additionally, we did not consider the influence of other comorbidities such as obesity and diabetes. Finally, our sample size was relatively small, though statistical analysis confirmed it was sufficient to ensure 90% power and a significance level of 0.05. We used G Power 3.1.9.7 software, adopting the following parameters: a point-biserial correlation and a *t*-test, with an effect size of 0.3, an alpha of 0.05, and a power of 0.9, which resulted in a sample size of *n* = 107 patients.

## 5. Conclusions

In conclusion, this study established a functional profile of patients with severe KOA with an indication for TKR consisting of self-reported questionnaires (VAS, WOMAC, and SF-36) and functional parameters (knee ROM, thigh circumference, maximum voluntary isometric contraction, and 6 min walk test). We identified quadriceps strength and 6MWT as the two most significant functional parameters associated with KOA severity, providing valuable information for guiding TKR decisions.

## Figures and Tables

**Figure 1 jfmk-09-00216-f001:**
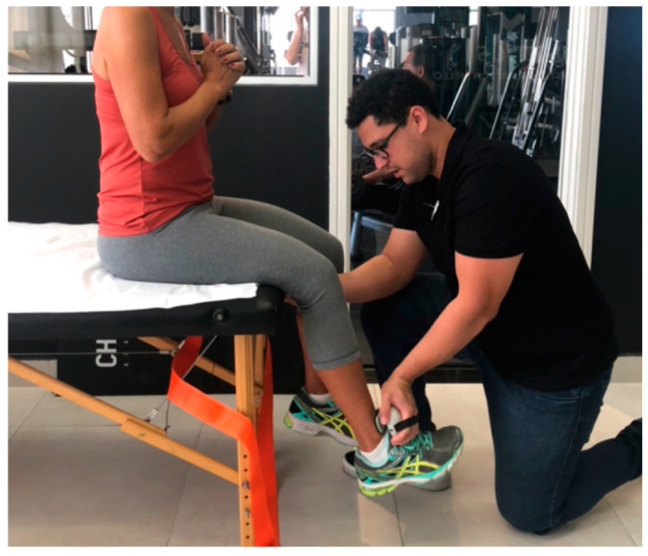
Position for collecting quadriceps strength.

**Figure 2 jfmk-09-00216-f002:**
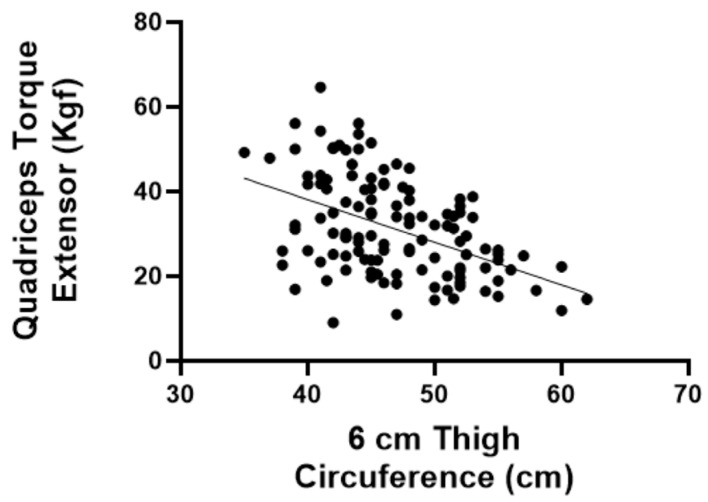
Spearman’s coefficient for correlation between quadriceps torque and thigh circumference 6 cm above the patella.

**Figure 3 jfmk-09-00216-f003:**
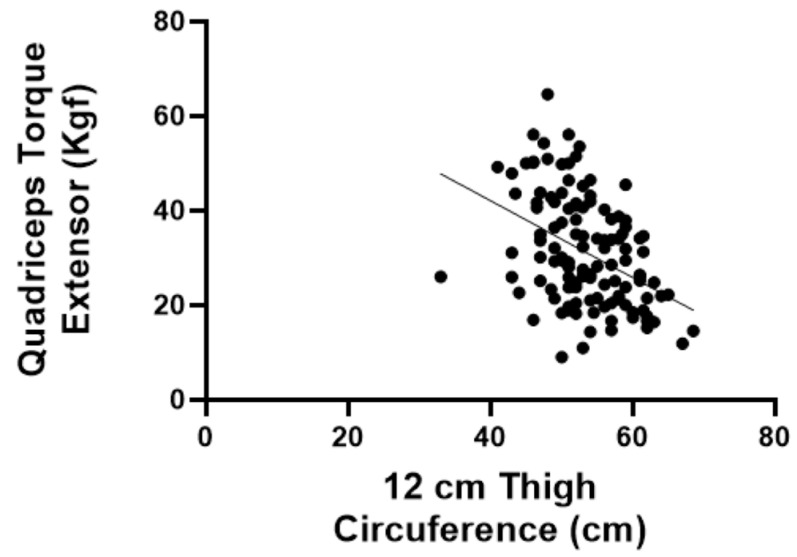
Spearman’s coefficient for correlation between quadriceps torque and thigh circumference 12 cm above the patella.

**Table 1 jfmk-09-00216-t001:** Patient demographics.

Demographics	Mean	SD
Age (years)	68	8
Weight (kg)	79.6	15.0
Size (cm)	164.2	8.5
BMI (kgm^2^)	29.4	4.5
Duration of symptoms (months)	52	50

SD = standard deviation; kg = kilograms; cm = centimeters; kgm^2^ = kilograms per square meter.

**Table 2 jfmk-09-00216-t002:** Patient functional characteristics.

Functional Characteristics	Index Knee	SD	Contralateral Knee	SD
Flexion (°)	121.1	12.3	130.4	1.3
Extension (°)	−4.9	6.0	−1.3	4.5
6 cm circumference (cm)	46.0	5.3	46.8	5.4
12 cm circumference (cm)	52.5	5.8	53.3	5.9
Extension torque (kgf)	19.1	8.0	24.3	8.7

cm = centimeter, kgf = kilogram force.

**Table 3 jfmk-09-00216-t003:** Six-minute walk test results.

6MWT Results	Mean	SD
6MWT (meters)	318.2	134.1
6MWT deficit (%)	−26.64	30.77

**Table 4 jfmk-09-00216-t004:** Average scores of patients on self-reported questionnaires.

Scores	Mean	SD
WOMAC	46.1	17.7
VAS	4.7	2.8
SF-36 MH	62.9	21.4
SF-36 EA	51.1	39.4
SF-36 SA	62.9	26.1
SF-36 V	54.8	23.1
SF-36 OHS	61.8	23.4
SF-36 PAIN	40.3	22.4
SF-36 PL	32.1	33.3
SF-36 FC	29.5	20.2

SD = Standard Deviation; WOMAC = Western Ontario and McMaster Universities Osteoarthritis Index; VAS = Visual Analog Scale; Short Form Health Survey = 36 subscales: MH = Mental health; EA = Emotional Aspects; SA = Social Aspects; V = Vitality; OHS = Overall Health Status; PL = Physical limitations; FC = Functional Capacity.

**Table 5 jfmk-09-00216-t005:** Spearman’s coefficient for correlation between functional aspects and self-reported variables.

Variables	WOMAC	VAS	MH	EA	SA	V	OHS	Pain	PL	FC
Flexion (°)	** −0.32 * **	−0.24 *	0.09	0.04	0.06	0.02	−0.12	0.10	0.14	0.13
Extension (°)	0.08	0.22	−0.01	−0.00	−0.08	0.06	0.02	−0.05	−0.13	−0.09
6 cm Thigh Circumference (cm)	0.20 *	0.21 *	−0.04	−0.10	−0.05	−0.13	0.04	−0.25 *	−0.20 *	−0.24 *
12 cm Thigh Circumference (cm)	0.19 *	0.23 *	0.00	−0.99	−0.03	0.10	0.07	−0.26 *	−0.19 *	−0.21 *
Extensor Torque (kgf)	** −0.31 * **	** −0.32 * **	0.23 *	0.14	0.18 *	0.24 *	0.18 *	0.12	0.10	** 0.32 * **
6MWT	** −0.35 * **	** −0.48 * **	** 0.34 * **	0.20 *	0.29 *	** 0.32 * **	0.19 *	0.23 *	0.13	** 0.44 * **
6MWT deficit	−0.29 *	−0.29 *	** 0.32 * **	0.20 *	0.27 *	** 0.33 * **	0.22 *	0.16	0.07	** 0.34 * **

6MWT deficit = difference between the minimal distance walked within the 6MWT. MH = Mental health; EA = Emotional Aspects; SA = Social Aspects; V = Vitality; OHS = Overall Health Status; PL = Physical limitations; FC = Functional Capacity. * Correlations with *p* < 0.05; **Bold and underline: moderate correlation**.

## Data Availability

The data presented in this study are available on request from the corresponding author.

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
