# Peer review of "A Complete Functional Characterization of Patients with Severe Knee Osteoarthritis in Need of Total Knee Replacement"

_jfmk, 2024, doi:10.3390/jfmk9040216_

Round 1

Reviewer 1 Report

Comments and Suggestions for Authors

Overall, the study presents valuable insights into functional parameters and their correlation with knee osteoarthritis (KOA) severity. However, there are several areas like methods and discussion that require more clarity . I hope you find my comments helpful. Some specific items for clarification are outlined below:

·         Abstract: Provide a clear statement in background why this study is important.

·         Abstract: The objective statement is too wordy and doesn't concisely state what the study aims to achieve. Make it more direct and clearly state the specific goal

·         Abstract: In results provide enough specific data or statistics (e.g., correlation coefficients, p-values).

·         Abstract: The conclusion should emphasize how the findings could be used in clinical practice or future research.

·         Introduction: There are several instances of redundancy, such as repeating the global prevalence of KOA and general statements about functional impairment. Some parts could be condensed for clarity. E.g.: Statement can be reduced as "Osteoarthritis (OA) affects approximately 250 million people globally, with the knee being the most commonly affected joint."

·         Introduction: Clearly state the objective and highlight the gap in the literature this study aims to address.

·         Introduction: The introduction does not sufficiently highlight the knowledge gap or the clinical importance of identifying functional parameters in patients before TKR. Emphasize why it's crucial to study the relationship between functional tests and outcomes in KOA, linking it to the problem of patient dissatisfaction post-TKR.

·         Methods: While the study is described as "cross-sectional," there is no clear explanation of the timeline, recruitment process, or setting for patient selection. Clarify how patients were recruited (e.g., consecutive sampling from a waiting list), the recruitment period, and how data collection was conducted (e.g., whether assessments were conducted during routine clinic visits).

·         Methods: Explicitly state that informed consent was obtained from all participants and briefly describe how patient confidentiality was ensured.

·         Methods: it’s unclear how the severity of KOA was determined before enrolling patients. Be more specific about the inclusion criteria. For instance, clarify what level of pain qualifies a patient or specify the duration of pain. Clearly define how the "previous indication for knee arthroplasty" was determined (e.g., by a surgeon).

·         Methods: The exclusion criteria mention "patients with another disease not correlated with KOA that could interfere with the performance of the functional tests," but it doesn’t specify what kinds of conditions were excluded or how they were assessed. Please elaborate

·         Methods: Provide more specific information on how each functional testing procedure was administered, including instructions given to patients, and how results were recorded.

·         Methods: Mention the validity of the tools used and whether reliability measures (e.g., inter-rater reliability for radiographic assessment) were considered.

·         Methods: The statistical analysis section does not specify how missing data were managed. Include a brief statement on how missing data were managed

·         Methods: There is no mention of a sample size calculation or power analysis, which is important to assess whether the study was sufficiently powered to detect meaningful correlations. whether a power calculation was performed and how the sample size was determined.

·         Methods: Clarify whether the examiners performing functional tests or collecting self-reported data were also blinded to other patient information.

·         Results: Provide more comprehensive results in text, including ranges or confidence intervals for the means and p-values for all comparisons.

·         Discussion: The discussion covers many points without a clear structure, making it difficult to follow the main argument. The reader may lose track of the primary takeaways due to the lack of a cohesive narrative. Organize the discussion in a structured way, separating sections such as:

·         Key findings and how they address the research question

·         Comparison with existing literature

·         Implications for clinical practice

·         Limitations

·         Suggestions for future research

·         Discussion: Avoid unnecessary repetition and instead focus on expanding the discussion with deeper insights into each finding. For example, if quadriceps strength is important, explain why in more depth or how it impacts treatment outcomes differently in various patient subgroups.

·         Discussion: the correlation between quadriceps strength and KOA symptoms is mentioned multiple times. Provide a more detailed interpretation of why these findings matter and how they contribute to the field. Discuss potential underlying mechanisms or clinical pathways that explain your results.

·         Discussion: Add limitation related to small sample size, context of generalizability.

·         Discussion: Expand on how the findings can directly inform clinical decision-making.

·         Discussion: Discuss the mental health aspect. how mental health interventions could be integrated into KOA treatment plans, particularly for those experiencing depression and anxiety related to their condition.

·         Discussion: The mention of "significance level of 0.05%" is incorrect. Correct the statistical terminology to "a significance level of 0.05."

·         Conclusion: Includes a more detailed wrap-up of the study’s scope. Emphasizes how the findings can impact clinical decision-making.

·         Please check for clarity and grammatical accuracy.

Comments on the Quality of English Language

The manuscript requires improvements in clarity, grammar, and conciseness to enhance readability.

Author Response

Dear reviewer,

Thank you so much for the revision. I've agreed to all your points and hope you accept the changes we've made.

 comment 1) Abstract: Provide a clear statement in background why this study is important.

We added ": Current literature lacks objective criteria to correctly identify patients in need of a total knee replacement. Surgery indication can be challenging for orthopedic surgeons, which may lead to high levels of patient dissatisfaction"

comment 2: Abstract: The objective statement is too wordy and doesn't concisely state what the study aims to achieve. Make it more direct and clearly state the specific goal

we made it more objective: "The objective of this study is to describe a complete set of functional characteristics to identify patients with end-stage knee osteoarthritis in need of a total knee replacement, correlating data of strength and performance tests with pain, function and quality of life questionnaires"

comment 3: Abstract: In results provide enough specific data or statistics (e.g., correlation coefficients, p-values).

We added it

comment 4: Abstract: The conclusion should emphasize how the findings could be used in clinical practice or future research.

we changed to "The present article established a functional profile of patients with severe knee osteoarthritis with indication for total knee replacement, which may help orthopedic surgeons in their decision process. "

comment 5:    Introduction: There are several instances of redundancy, such as repeating the global prevalence of KOA and general statements about functional impairment. Some parts could be condensed for clarity. E.g.: Statement can be reduced as "Osteoarthritis (OA) affects approximately 250 million people globally, with the knee being the most commonly affected joint."

OK

comment 6: Introduction: Clearly state the objective and highlight the gap in the literature this study aims to address.

"Moreover, current literature poorly correlates preoperative factors for the decision to perform surgery with clinical outcomes and level of patient satisfaction[11,13]"

comment 7: Introduction: The introduction does not sufficiently highlight the knowledge gap or the clinical importance of identifying functional parameters in patients before TKR. Emphasize why it's crucial to study the relationship between functional tests and outcomes in KOA, linking it to the problem of patient dissatisfaction post-TKR.

Done

comment 8: ·         Methods: Explicitly state that informed consent was obtained from all participants and briefly describe how patient confidentiality was ensured.

"The objectives, benefits, and risks of the study were explained to all patients. After fully understanding the study and its implications, patients who agreed to participate signed a free and informed consent form. All data linked to the patients were stored using only the hospital's identification code, with no personal names or photos associated with the participants."

comment 9: Methods: it’s unclear how the severity of KOA was determined before enrolling patients. Be more specific about the inclusion criteria. For instance, clarify what level of pain qualifies a patient or specify the duration of pain. Clearly define how the "previous indication for knee arthroplasty" was determined (e.g., by a surgeon).

Inclusion criteria were (i) knee pain for at least year with VAS>3; (ii) radiographic knee osteoarthritis Kellgren and Lawrence scheme > 2; (iii) patients in a waiting list for a total knee replacement at the Knee Surgery outpatient clinic of a University Orthopedics service (indication by a experienced surgeon). 

comment 10: Methods: The exclusion criteria mention "patients with another disease not correlated with KOA that could interfere with the performance of the functional tests," but it doesn’t specify what kinds of conditions were excluded or how they were assessed. Please elaborate

We excluded patients with signs of knee joint infection, pregnancy, patients with another disease not correlated with KOA that could interfere with the performance of the functional tests, such as severe respiratory disease or lower limb palsy.  

comment 11: Methods: Provide more specific information on how each functional testing procedure was administered, including instructions given to patients, and how results were recorded.

Done

comment 12: Methods: Mention the validity of the tools used and whether reliability measures (e.g., inter-rater reliability for radiographic assessment) were considered.

Two of the authors (GCC, ARZ) examined all radiographs to classify the severity of OA using the Kellgren and Lawrence scheme[18]. In cases with interobserver disagreement a third opinion would be consulted. Observers were blind to functional results.

Functional tests and questionnaires were applied by only one investigator

comment 13 Methods: The statistical analysis section does not specify how missing data were managed. Include a brief statement on how missing data were managed

added at 

2.5.Statistical analysis:

"Only patients with complete data were to be included in the final analysis. Patients with missing data were excluded."

comment 14: 

 Methods: There is no mention of a sample size calculation or power analysis, which is important to assess whether the study was sufficiently powered to detect meaningful correlations. whether a power calculation was performed and how the sample size was determined.

"No sample size calculation was performed prior to the study, since we recruited all individuals of a wating list. Nevertheless, a power calculation was performed to ensure that our sample size was adequate"

To calculate the sample size for our study, we used G Power 3.1.9.7 software, adopting the following parameters: point-biserial correlation and a t-test, with an effect size of 0.3, an alpha of 0.05, and a power of 0.9, which resulted in a sample size of n=107 patients.

comment 15:     Methods: Clarify whether the examiners performing functional tests or collecting self-reported data were also blinded to other patient information.

There was no blinding

comment 16:  Results: Provide more comprehensive results in text, including ranges or confidence intervals for the means and p-values for all comparisons.

Done

comment 17: 

 Discussion: The discussion covers many points without a clear structure, making it difficult to follow the main argument. The reader may lose track of the primary takeaways due to the lack of a cohesive narrative. Organize the discussion in a structured way, separating sections such as:

  • Key findings and how they address the research question
  • Comparison with existing literature
  • Implications for clinical practice
  • Limitations
  • Suggestions for future research

OK

comment 18: Discussion: the correlation between quadriceps strength and KOA symptoms is mentioned multiple times. Provide a more detailed interpretation of why these findings matter and how they contribute to the field. Discuss potential underlying mechanisms or clinical pathways that explain your results.

" A recent systematic review and meta-analysis that included 46819 men and women confirmed knee extensor muscle weakness as a risk factor for the development of KOA[23]. The main explanation for this association is that the quadriceps muscle regulate knee joint loading and motion, which is critical for cartilage homeostasis. A week quadriceps can negatively influence loading distribution, leading to degenerative joint pathology[23]. "

comment 19: Discussion: Add limitation related to small sample size, context of generalizability.

Finally, our sample could be considered small, which may prevent the finding from being extrapolated. We recruited all patients in the waiting list for TKA in our service. But our statistical analysis did show that this sample was sufficient to ensure a power of 90% and a level of significance of 0.05 for all analyses. We used G Power 3.1.9.7 software, adopting the following parameters: point-biserial correlation and a t-test, with an effect size of 0.3, an alpha of 0.05, and a power of 0.9, which resulted in a sample size of n=107 patients.

comment 20: Discussion: Discuss the mental health aspect. how mental health interventions could be integrated into KOA treatment plans, particularly for those experiencing depression and anxiety related to their condition.

We also found low scores on the emotional aspects and mental health subscales of SF-36 questionnaire. Mental health disorders are a common issue in KOA patients. A recent systematic review and meta-analysis examined 14 studies involving adult patients with KOA and found a significant prevalence of depression and anxiety among older KOA patients. They found a pooled prevalence of depression of 30%, and a combined prevalence of anxiety of 27%[28]. Persistent pain and mobility restrictions surely play a significant role in the development of such mental health conditions. Mental health interventions should definitely be integrated into KOA treatment plans. More studies are needed in that area"

comment 21: Conclusion: Includes a more detailed wrap-up of the study’s scope. Emphasizes how the findings can impact clinical decision-making. 

"

In conclusion, the present article established a functional profile of patients with severe KOA with indication for total knee replacement consisting of self-reported questionnaires (VAS, WOMAC and SF-36) and functional parameters (knee range of motion, thigh perimeter measurement, maximum voluntary isometric contraction and 6- minute walk test). We also identified quadriceps strength and 6MWT as the two most important functional parameters that correlate with KOA severity. This finding provide more information to help the decision making process of a total knee replacement indication."

Reviewer 2 Report

Comments and Suggestions for Authors

The authors describe a set of self-reported and performance-based functional characteristics and pain to identify severe knee osteoarthritis (KOA) patients in need of total replacement knee surgery. Interestingly, quadriceps strength and 6-minute walk test were identified as two most important functional parameters that correlate with KOA severity. This finding is interesting. However, the authors brought up in the introduction that a good patient selection is crucial for successful TKR. It remains to be seen whether patient satisfaction will be increased with these criteria. In a recent study by Rutter-Locher et al. (PAIN 165, 2578-2585, 2024) performed a randomised controlled trial of the effect of intra-articular lidocaine on pain in inflammatory arthritis. The authors found that that painDetect score predicted persistent high pain after intra-articular lidocaine. Earlier data shows that KOA patients with neuropathic pain symptoms suffer from more severe pain. Further, a systematic review and meta-analysis suggests that KOA patients with neuropathic symptoms may not benefit from knee replacement surgery as much as patients with peripherally-driven nociceptive pain (Wluka et al. Osteoarthritis and Cartilage,28,1403-1411 2020). It would have been interesting to know whether subset of patients in the present study show also persistent pain assessed by painDetect score and is there a correlation with painDetect score with quadriceps strength and 6-minute walk test and are there differences in patient satisfaction after TKR surgery at 6 or 12 months post-surgery. Please discuss.

Author Response

Dear reviewer,

Thank you so much for your comment. We totally agree with you.

Unfortunately we did not look to psychological factors in this study, just functional aspects. It is certainly something that we should include in our patient "profile"

we added this to the discussion:

"We also found low scores on the emotional aspects and mental health subscales of SF-36 questionnaire. Mental health disorders are a common issue in KOA patients. A recent systematic review and meta-analysis examined 14 studies involving adult patients with KOA and found a significant prevalence of depression and anxiety among older KOA patients. They found a pooled prevalence of depression of 30%, and a combined prevalence of anxiety of 27%[28]. Persistent pain and mobility restrictions surely play a significant role in the development of such mental health conditions. On the other hand, it has been shown that patients with higher painDETECT scores (a questionnaire used to assess the presence of central pain) presented with persistent high pain after treatment[29]. Thus, this could be also used as a screening tool to identify patients that could be dissatisfied after a TKR. Mental health interventions should definitely be integrated into KOA treatment plans. More studies are needed in that area"

Reviewer 3 Report

Comments and Suggestions for Authors

This paper studies pain and function in end-stage knee osteoarthritis patients needing surgery, correlating physical tests with questionnaires.

But there are still the following terms that can be improved:

1.  It is suggested to further explore the validity and limitations of using thigh circumference as an indirect indicator, especially considering its applicability to patients with different body types.

2. When discussing treatment options, it is suggested to elaborate on the differences in effectiveness of surgical alternatives, particularly highlighting the latest advances in conservative treatments.

3.  It is recommended to mention whether comorbid factors such as obesity and diabetes were considered in the analysis. 

Author Response

Dear reviewer,

Thank you very much for your comments.

We agree with it all

Regarding the suggestion to further explore the validity and limitations of using thigh circumference as an indirect indicator, especially considering its applicability to patients with different body types, we added to the discussion:

"Another important finding was the correlation between thigh circumference (at 6 cm and 12 cm) and extensor torque. We found that extensor torque had a significant positive correlation with thigh circumferences at 6 cm (r = -0.4372; p < 0.05) and 12 cm (r = -0.4144; p < 0.05). Thigh circumference is a reliable measurement of muscular mass and can be used as a parameter to identify sarcopenia, independent of the body mass composition or patient's fat distribution[27]. Dynamometers are expensive devices and may not be universally available, specially in developing countries like Brazil. Therefore, if it is impossible to measure maximum strength due to the unavailability of a dynamometer, thigh circumference could work as a surrogate outcome, since it is easy to perform and virtually inexpensive."

We made it more clear that unfurtunatelly our analysis did not include any influence of comorbidities

We also made other changes proposed by the other reviewers and hope our paper is now fit to publication

Thank you again!

Round 2

Reviewer 1 Report

Comments and Suggestions for Authors

Thank you for incorporating suggested comments. 

Comments on the Quality of English Language

NA

Author Response

Dear reviewer,

Thank you!